# RIAIT (Italian Registry of Allergen Immunotherapy): Protocol for a New Tool in a New Vision of Disease-Modifying Therapy for Allergists

**DOI:** 10.3390/jpm14080854

**Published:** 2024-08-12

**Authors:** Giovanni Costanzo, Cristiano Caruso, Giovanni Paoletti, Ilaria Baglivo, Stefania Colantuono, Diego Bagnasco, Marco Caminati, Mattia Giovannini, Riccardo Castagnoli, Gianenrico Senna, Concetta Sirena, Maria Angela Tosca, Giovanni Passalacqua, Gian Luigi Marseglia, Michele Miraglia del Giudice, Giorgio Ciprandi, Cristiana Indolfi, Salvatore Barberi, Massimo Landi, Mario Di Gioacchino, Giorgio Walter Canonica, Enrico Heffler

**Affiliations:** 1Personalized Medicine, Asthma and Allergy, IRCCS Humanitas Research Hospital, 20089 Rozzano, Italy; giovanni.costanzo@humanitas.it (G.C.); giovanni.paoletti@hunimed.eu (G.P.); giorgio_walter.canonica@hunimed.eu (G.W.C.); 2UOSD Allergy and Clinical Immunology Unit, Fondazione Policlinico A. Gemelli, IRCCS, 00168 Rome, Italy; cristiano.caruso@policlinicogemelli.it; 3Department of Translational Medicine and Surgery, Università Cattolica del Sacro Cuore, 00168 Roma, Italy; 4Department of Biomedical Sciences, Humanitas University, 20072 Pieve Emanuele, Italy; 5UOC Digestive Disease Center CEMAD, Fondazione Policlinico A. Gemelli, IRCCS, 00168 Rome, Italy; ilaria.baglivo@guest.policlinicogemelli.it; 6UOSD DH Internal Medicine and Digestive Disease, Fondazione Policlinico A. Gemelli, IRCCS, 00168 Rome, Italy; stefania_colantuono@guest.policlinicogemelli.it; 7UO Clinica Malattie Respiratorie e Allergologia, IRCCS-AOU San Martino, 16132 Genova, Italy; diego.bagnasco@dimi.unige.it (D.B.); passalacqua@unige.it (G.P.); 8Dipartimento di medicina interna (DIMI), Università degli Studi di Genova, 16146 Genova, Italy; 9Allergy Unit and Asthma Center, Verona Integrated University Hospital, 37134 Verona, Italy; marco.caminati@univr.it (M.C.); gianenrico.senna@aovr.veneto.it (G.S.); 10Department of Medicine, University of Verona, 37124 Verona, Italy; 11Allergy Unit, Meyer Children’s Hospital IRCCS, 50139 Florence, Italy; mattia.giovannini@unifi.it; 12Department of Health Sciences, University of Florence, 50139 Florence, Italy; 13Pediatric Unit, Department of Clinical, Surgical, Diagnostic and Pediatric Sciences, University of Pavia, 27100 Pavia, Italy; riccardo.castagnoli@unipv.it (R.C.); gl.marseglia@smatteo.pv.it (G.L.M.); 14Pediatric Clinic, Fondazione IRCCS Policlinico San Matteo, 27100 Pavia, Italy; 15Registro Italiano Allergen Immunotherapy (RIAIT), Via San Gregorio 12, 20124 Milan, Italy; 16Allergy Center, IRCCS Istituto Giannina Gaslini, 16147 Genoa, Italy; mariangelatosca@gaslini.org; 17Department of Woman, Child and General and Specialized Surgery, University of Campania “Luigi Vanvitelli”, 80138 Naples, Italy; michele.miraglia@unina2.it (M.M.d.G.); cristianaind@hotmail.com (C.I.); 18Allergy Clinic, Casa di Cura Villa Montallegro, 16145 Genoa, Italy; gio.cip@libero.it; 19Pediatric Unit, ASST-Rhodense, Rho, 20024 Milan, Italy; salvatorebarberi@hotmail.com; 20Department of Medical Sciences, Graduate School of Allergology and Clinical Immunology, University of Turin, 10126 Turin, Italy; landi@alma.it; 21Institute for Clinical Immunotherapy and Advanced Biology Treatments, University of Chieti, 66100 Chieti, Italy; m.digioacchino@unich.it

**Keywords:** allergy, immunotherapy, real world, registry, biomarker

## Abstract

Randomized controlled trials have demonstrated responses to clinical parameters, but a significant proportion of allergy patients in real-life settings would have been excluded from such studies. Therefore, real-world research is needed, and there is a growing body of information on allergen immunotherapy’s long-term effectiveness and safety. Real-world evidence can be a valuable instrument to better understand the patient’s journey and the effectiveness and safety of therapies. For this purpose, a registry will be used for the first time in Italy to evaluate the impact of allergen immunotherapy on several outcomes, including quality of life and disease-related effects in the pediatric and adult allergic population with a socio-economic assessment and respect to real-world health.

## 1. Background

Allergen immunotherapy (AIT) consists of administering increasing amounts of a specific allergen, to which the patient is allergic, at regular intervals for a long time (usually three to five years) to modulate the response of the patient’s immune system to the same allergen [1,2,3]. Although novel applications and delivery strategies are regularly proposed, AIT is currently used to treat allergic respiratory diseases (allergic rhinitis and allergic asthma), food allergy, and insect venom allergy [3,4]. For the scope of this article, regarding the term AIT, we will refer to allergic rhinitis and allergic asthma treatment only; this therapy is offered to individuals who are affected by immunoglobulin E (IgE)-mediated allergic rhinitis and/or allergic asthma due to inhaled allergens, such as pollens, house dust mites (HDMs), molds, or pet danders.

Unlike conventional pharmacotherapeutic approaches (e.g., antihistamines, leukotriene-receptor antagonists, and inhaled and systemic corticosteroids), which act to reduce/control signs and symptoms, AIT can be considered the only disease-modifying treatment in the aforementioned diseases, as it has been shown to modify the natural evolution of type 2 IgE-mediated diseases and to provide long-lasting effect in terms of responsiveness to selected allergens [5,6,7]. In fact, AIT acts on different aspects of the immune response: it reduces IgE-mediated mast cell degranulation, induces Treg cell activation, and promotes the production of allergen-specific IgG4 antibodies that block the IgE activation mediating the inflammatory cascade [7,8]. Moreover, especially in children and young people, AIT might reduce the risk of developing clinical manifestations of asthma and of using asthma medication in the future, arresting the so-called “allergic march” [9,10].

The two main administration routes are subcutaneous immunotherapy (SCIT) and sublingual immunotherapy (SLIT). Both have been proven effective and safe, and each administration route has pros and cons [11].

Interestingly, AIT has been used for over a century, when in 1911 Leonard Noon was the first to show that repeated injections of crude grass pollen extract into individuals with hay fever reduced allergic sign and symptoms to the exposition of the grass pollen [12], a strategy perhaps conceived in analogy with prophylactic vaccination using killed or modified causal organisms for the prevention of infectious diseases; in fact, the term “pollen vaccination” is sometimes used to inaccurately yet effectively indicate AIT.

Over the years, AIT has collected crescent evidence on the efficacy [1,2], safety [13,14,15,16,17], short-term benefits, and long-term benefits, years after its discontinuation [15,18,19]; these achievements have led to important international recognition, with two milestones: the 1998 position paper by the World Health Organization (WHO) [1] and the 2013 World Allergy Organization (WAO) position paper [2]. Nevertheless, an urgent need for quality real-world data has arisen in the same documents.

Numerous random controlled trials (RCTs) have been conducted during the history of this more-than-century-old therapy, and they have provided the necessary evidence for the registration of several AIT products. Indeed, currently, AIT is cited in the most significant and updated international recommendations on the management of allergic rhinitis [20] and asthma [21,22].

Nonetheless, it is well known that RCTs are inherently difficult to conduct, and the ones that evaluate AIT are even more so due to the long span of therapy, the heterogeneity of patients, and the lack of reliable biomarkers [23].

As advocated by the Respiratory Effectiveness Group [24], the RCTs and real-life data such as registries are two equally essential methods and should be used complementarily in defining the effect of an intervention to evaluate the effectiveness of a given therapeutic intervention [25].

This article describes a protocol for a prospective multicenter registry for adults and children starting AIT. The existence of a specific, rigorous, and well-conducted registry would provide high-quality robust data to evaluate the short and long-term effectiveness of this therapy.

Moreover, there is a crucial and urgent need to create a standardized national dataset to facilitate interoperability, data sharing, and cross-comparison among centers. This mass of information might gain sufficient statistical power to address important clinical and research questions, and a national registry will minimize the variability of data collected by standardizing variables across geographical regions, thus enhancing our understanding of the allergic population by examining the response to AIT based on characteristics like phenotypes, biomarkers, concomitant treatments, and socio-economic status. Finally, long-term patient follow-up will enable a real-life understanding of respiratory allergic disease evolution when treated with AIT.

## 2. Methods

### 2.1. Registry Design and Governance

The Italian Registry of Allergy Immunotherapy (RIAIT) is a multicenter observational study that aims to collect the most extensive data available on Italian patients using AIT for respiratory allergies (rhinoconjunctivitis and/or asthma) in a real-life context, through a network of allergy reference centers with specific interest and expertise in the management of allergic respiratory patients.

Following this objective, the registry’s network aims to build an extensive database of patients treated with AIT, gaining the opportunity to collect Italian data and aiming to monitor the evolution of the disease over time during undergoing AIT.

The registry is promoted by the Italian Society of Allergy, Asthma and Clinical Immunology (SIAAIC) [26] and the Italian Society of Pediatric Allergy and Clinical Immunology (SIAIP) [27].

The Italian Society of Allergy, Asthma and Clinical Immunology (SIAAIC) is the biggest Italian scientific society of allergy and clinical immunology; founded in 1953, it is strongly represented on the Italian national territory with over 1700 active members and various thematic areas of interest such as respiratory diseases, dermatologic diseases, allergology, and clinical immunology. The Italian Society of Pediatric Allergy and Clinical Immunology (SIAIP) was founded in 1997 and aims to increase the spread of allergological and immunological culture, not only in the scientific world, but also in society in general, taking into account the negative burden that allergic diseases often have on the health and quality of life of children and their families.

### 2.2. Study Design

This is a prospective multicenter observational registry of patients suffering from conjunctivitis, rhinitis, and/or allergic asthma eligible for and treated with AIT according to EAACI (European Allergy and Asthma Immunology) Guidelines [21].

#### 2.2.1. Inclusion and Exclusion Criteria

The registry will enroll consecutive patients who must meet all the following inclusion criteria:Age ≥ 1 year.Confirmed diagnosis of allergic rhinitis, conjunctivitis, and/or allergic asthma according to EAACI 2019 guidelines [21]. The confirmation of allergenic sensitization must be documented by skin prick tests and/or serum specific IgE.Having been followed by the recruiting center for at least one month. This is considered necessary to establish that the patients are genuinely affected by allergic respiratory disease and are eligible for AIT; in this period, screening tests are carried out to exclude and/or confirm concomitant diseases (comorbidities). In fact, according to the aforementioned EAACI guidelines [21], a patient can be defined as genuinely eligible for AIT only after a preliminary phase in which the patient is treated with a therapy targeted to the clinical manifestations. This observation period allows a correct evaluation of the real adherence and response to therapy, as well as the identification and appropriate treatment of any comorbidity and the possible elimination of aggravating factors.Having received the prescription of any of the available AIT products in Italy, irrespectively of the selected allergens and the administration route and schedule.

There will be no exclusion criteria to allow a real-life vision of the characteristics of these patients eligible for or treated with AIT.

#### 2.2.2. Data Collection

Patients’ available data, including anamnestic, clinical, radiological, and bio-humoral characteristics, will be collected at enrolment and every 12 months thereafter, for a maximum follow-up period of 10 years, through an electronic case report form (eCRF) developed with the REDCap^®^ (Research Electronic Data Capture) software (v14.1.2).

The eCRF compilation can also occur through a Uniform Resource Locator (URL) generated through the REDCap software.

Variables collected at the baseline and every 12-month follow-ups include the following:○Anthropometric characteristics (age, height, gender, and body mass index BMI).○Clinical characteristics (presence of rhinitis and/or conjunctivitis and/or asthma; allergen sensitizations assessed by means of skin prick tests and/or serum specific IgE; presence of comorbidities; lung function parameters; need for hospitalizations and/or visits to the emergency room in recent years, etc.).○Patients’ reported outcomes (PROs): e.g., Rhinoconjunctivitis Quality of Life Questionnaire (RQLQ) [28], Pediatric Rhinoconjunctivitis Quality of Life Questionnaire (PRQLQ) [29]; level of control of rhinitis and/or asthma in the last month, according to Allergic Rhinitis and its Impact on Asthma (ARIA) guidelines [30] and with standardized questionnaires, such as the Total Nasal-Symptom Score (TNSS) [31], the Asthma Control Test Asthma (ACT) [32], and the Asthma Control Questionnaire (ACQ) [33]; the RhinAsthma Patient Perspective (RAPP) [34]; the Asthma Quality of Life Questionnaire (AQLQ) [35]; and the Pediatric Asthma Quality of Life Questionnaire (PAQLQ) [36].○Dosage and posology pharmacological therapies for allergic rhinoconjunctivitis and/or asthma.○Dosages and posology of drugs used for treating comorbidities (e.g., rhinosinusitis; Gastroesophageal Reflux Disease, GERD; atopic dermatitis, etc.).○Allergen immunotherapy details: type and name of the product, given allergens, route of administration, posology, and dosage.○Report and classification of adverse reactions during treatment with traditional drugs and/or AIT.○When available, indirect biomarkers of type-2 inflammation, e.g., as blood eosinophils, serum IgE, and fractional exhaled nitric oxide (FeNO); and lung function parameters.○Adverse events and any reasons for any interruption or switch of AIT.

Any further tests aimed at better phenotyping allergic rhinitis and/or bronchial asthma will be taken into consideration based on the availability of each recruiting center. No genetic investigations or other investigations are planned at the moment.

#### 2.2.3. Objectives

Our objectives include the following:○Evaluate the short- and long-term real-life effectiveness and safety of AIT overall, in specific patient groups’ phenotypes and with specific AIT products.○Evaluate the pertinence and differences in suggesting AIT to a patient, comparing the physician’s evaluation in various settings with established international guidelines, describing the factors associated with treatment choices and changing over time, and promoting interoperability, data sharing, and cross-comparison among centers.○Describe long-term respiratory allergic disease evolution in patients treated with AIT, especially in the pediatric population.○Describe the natural history of the patient population with respiratory allergies and identify patient groups describing their illness burden, management patterns, and clinical progression.○Promote the creation of accurate, standardized, and efficient processes for diagnosing and treating respiratory allergic diseases, especially with AIT.○Assess biomarker data to predict diagnosis, treatment response, and long-term disease progression.

#### 2.2.4. Pharmacovigilance

The data collected also regards the frequency and grading of immediate and delayed and local and systemic adverse reactions attributable to the administration of AIT, as well as the resulting therapeutic management of the events.

#### 2.2.5. Duration of the Study and Statistics

The aforementioned data will be collected for 10 years for each patient. Additionally, periodic statistical analyses will be conducted by the study’s central data manager. The main target will be describing the enrollment trend in the various centers, evaluating the completeness of the information collected, and quality control. The following register comprehensiveness indicators for each center will include the following: the number of patients recruited, the percentage of patients recruited compared to the expected target, and the percentage of missing data per variable. Moreover, descriptive analyses will be conducted to display the distribution of the main characteristics of patients by each center (e.g., sex, age, comorbidities, allergen sensitization profile, the severity of the disease, outcomes, ongoing therapies, etc.).

Quantitative variables will be expressed as mean and standard deviation (or median and interquartile range if distributed asymmetrically). Qualitative variables will be expressed as absolute frequencies and percentages.

The calculation of statistical power or precision is calculated on the basis of the existing sample size. The normality of the continuous variables will be evaluated with the Kolmogorov–Smirnov test. Differences between groups among qualitative variables will be analyzed with the chi-square test or Fisher’s exact test depending on their distribution. Differences between groups among quantitative variables are analyzed via Student’s t-test or Mann–Whitney, on the basis of their distribution.

A two-tailed *p*-value with values inferior to 0.05 will be considered significant. Multivariate or univariate linear regression analyses will be performed among the variables based on a 95% confidence interval. Advanced statistical models, such as the Generalized Estimating Equation (GEE) or the Generalized Linear Model (GLM), will be used to identify responders and non-responders.

A medical data review will also be carried out once a year. These analyses allow for the identification of any corrective interventions aimed at improving the quantity and quality of the information collected among those provided in the CRF; these suggestions will be communicated to the manager of the individual center. The results of the analyses will be published anonymously (as aggregated data) in annual reports, which will be made available to the Principal Investigators of the centers and investigators involved in the project. The results of the analyses carried out will be the subject of scientific publications.

#### 2.2.6. Electronic Database, Data Management, and Confidentiality

The study data will be collected and managed via a specific eCRF accessible from the web. The eCRF is implemented by the REDCap^®^ software installed by the Italian Society of Asthma Allergology and Clinical Immunology (SIAAIC) and the Italian Society of Pediatric Allergy and Immunology (SIAIP). REDCap^®^ is a secure, web-based software platform created to support data collection for research purposes, providing the following: an intuitive user interface for validated data collection; audit trails to record any data manipulation and data export procedures; automatic export procedures in pseudo-anonymous format compatible with the most common statistical software; and data integration procedures and interoperability with external resources [37,38].

Each user of a center involved in the project receives an email with their username and must define their password. Through this “table-based” authentication method, the user passwords are stored in the database in an encrypted format using the SHA-512 hash function. Authentication is also secure thanks to an HTTP connection based on an SSL certificate. Through their credentials, each user can access a reserved area of the web application. The actions enabled for each user will depend on the administrator’s role associated with the user (data entry, data monitoring, etc.). The application generates a unique and pseudo-anonymous identification code for each patient. Each center maintains the enrollment list locally, based on the association between the identification data of the enrolled patient and the unique pseudo-anonymous code. No patient identification information will be stored within the database in accordance with the principle of data minimization and compliance with the general data protection regulation (GDPR, General Data Protection Regulation—EU Regulation 2016/679). The data ownership lies with the participating centers, SIAAIC and SIAIP.

#### 2.2.7. Ethics

The latest revision of the Helsinki Declaration, as well as the Oviedo Declaration, are the basis for the ethical conduct of the study. The study protocol is designed and will be conducted to ensure adherence to the principles and procedures of Good Clinical Practice and to comply with Italian laws, as described in the following documents and accepted, with their signature, by the study investigators:ICH Harmonized Tripartite Guidelines for Good Clinical Practice 1996.Directive 91/507/EEC, The Rules Governing Medicinal Products in the European Community.Legislative Decree n.211 of 24 June 2003.Legislative Decree n.200 of 6 November 2007.Ministerial Decree 21 December 2007.

The study protocol has been approved by the Central Ethics Committee (Comitato Etico Indipendente Humanitas, protocol number: 2732/2020, and subsequent amendments) and the enrollment in the other centers will start upon approval of each local Ethics Committee.

Together with the British Society for Allergy & Clinical Immunology (BSACI) Registry for Immunotherapy (BRIT) [39] and the international South-Eastern European Adverse Events Registry (ADER) [40,41], this is one of the first registries of patients who are eligible for and benefit from AIT that has been established and this will be a pilot project to be shared with other countries through the European Academy of Asthma Allergy and Clinical Immunology (EAACI) [42].

Inclusion criteria, observation time, and data collected are summarized in Figure 1.

## 3. Discussion

Allergen immunotherapy has been one of the cornerstones of the treatment of the most significant allergic respiratory diseases for years and it is cited in papers written by the major national and international societies that deal with recommendations and guidelines on the treatment of these pathologies, as can be seen in the EAACI guidelines on AIT in allergic rhinoconjunctivitis [20], Italian guidelines [30], and Global Initiative for Asthma (GINA) recommendations [22].

Nevertheless, in the same papers, various issues have been addressed, questioning the quality of the studies that eventually led to the approval of this therapy by organizations and emphasizing the differences in the various pharmaceutical products prescribed. Therefore, many open questions have been pointed out, and those unanswered queries remain controversial [43]. We think that the RIAIT registry might provide a substantial number of data, and therefore statistical power, sufficient to address these criticisms, as well as answer important research questions, standardize data, provide tools to deal with the unmet needs, and offer solid scientific foundations for future standards about AIT.

### 3.1. The Crescent Role of Real Data in Modern Medicine

In recent meta-analyses of thousands of different studies about AIT [44] and its use for AR [43] and allergic asthma [45], an essential divergence in some outcomes has been found, basically justified by the study design and different outcomes, study population, and products.

The RIAIT registry will provide many homogenous data on a vast and heterogeneous population, collecting similar, agreed, and consistent information. The openness of the inclusion criteria and the absence of exclusion criteria would lead to the acquisition of many combinations between products, patients, and disease sub-types, allowing the study of different drugs in different setups in a real-world scenario.

Moreover, to assist in the quality appraisal of observational comparative effectiveness research, in recent years a REal Life Evidence AssessmeNt Tool (RELEVANT) has been developed by the Respiratory Effectiveness Group (REG) and EAACI joint task force [46,47]; the tool was designed to identify evidence which is robust enough to inform clinical practice and to warrant consideration by guideline bodies and has been used to systematically review observational studies on the effectiveness of AIT in treating respiratory allergic diseases [48]. The study identified central defects of comparative effectiveness research on AIT and a general lack of high-quality life-effectiveness observational research, highlighting the need for robust and reliable real-life data.

The discussion on the possible use and the value of these data fits within the broader topic of the validation of real-life studies in the context of modern medicine. In a recent manifesto endorsed by other scientific organizations such as the World Allergy Organization and International Primary Care Respiratory Group [25], real-world (RW) studies, such as registries, are described as essential tools to assess the drugs’ “effectiveness” (defined as the actual effects in the RW/clinical practice) as opposed to randomized controlled clinical trials (RCTs) responsible for defining the drug’s “efficacy” (the effect in artificial conditions as randomized controlled clinical trials). For instance, in asthma, real-life studies have been showing results sometimes significantly divergent from the ones collected from RCT, due to differences between trial and real-life patients in many aspects crucial to predicting the treatment response (such as therapy adherence, body mass index, comorbidity, and race) [49,50,51,52]. Therefore, as promoted by the Respiratory Effectiveness Group, the RCTs and real-life research should be considered complementary [24,53]; registries possess the potential to determine whether results observed in RCTs can be applied to a broader and heterogenous population, or whether other hypotheses should be considered and tested (e.g., for instance, treatment effectiveness in subgroups of interest, safety in a wide range of populations, and cost-effectiveness in various healthcare systems).

The aforementioned statements refer to the broader debate on the “efficacy-effectiveness gap”, identified as the discrepancies between the outcomes reported in RCTs and those observed in real-world clinical practice [54,55,56,57]. Of course, RCTs remain the cornerstone of modern evidence-based medicine and are still considered the gold standard for high-quality research. Nonetheless, RTCs suffer from intrinsic limitations: firstly, strict inclusion criteria, which make the findings obtained challenging to apply to real-life and larger populations; secondly, a relatively short duration in time, generally insufficient when there is the will to understand the long-time effects of a therapy on a patient.

Registries present some limitations as well, especially if compared with RCTs; they do not provide randomization of patients, there might be missing or incomplete data, the enrolment of patients in registries is less supervised compared with RCT, and the follow-up presents a lower grade of standardization [58].

Considering the limitations and unique opportunities that this kind of study might provide, numerous registries have been created for severe asthma, designed to understand better the epidemiology, inflammatory profile, different phenotypes, and treatment characteristics. In Europe only, the Belgian Severe Asthma Registry (BSAR) [59], the German National Registry for Asthma (GAN) [60], and the Severe Asthma Network in Italy (SANI) [61] have been formed. Considering only the latter example, many insightful data have been collected during the years, becoming the foundation for a crescent number of articles on both clinical (such as the higher-than-predicted prevalence of bronchiectasis among severe asthmatics [62]) and pharmaco-economics aspects (such as the excessive use of oral corticosteroids [63] and its impact [64]), to name a few. These register-obtained data and the following scientific publications contributed to paying attention to the revealed criticalities, setting the trends for future corrections on the current guidelines, and suggesting original goals to deal with the novel real-world needs [65]. We expect that a similar initiative like the RIAIT registry might achieve the same success in delivering reliable and easily usable data in a real-world scenario [23].

To our knowledge, together with the BSACI Registry for Immunotherapy (BRIT) [39] and the international South-Eastern European Adverse Events Registry (ADER) [40,41], RIAIT is one of the first registries of patients using AIT. Keeping asthma as an inspiring example of the utility of registries, RIAIT could serve as a pilot project for further wider collaboration in analogy with the successful International Severe Asthma Registry (ISAR) [66].

### 3.2. Addressing Open Questions and Unmet Needs

Although AIT is a therapy considered safe in adults, children, and adolescents [17,40], and is effective in both allergic rhinitis [20] and asthma [45] with a large spectrum of indications, some relative and absolute contraindications on the use of AIT subsist. According to the EAACI respective guidelines on the two allergic respiratory diseases [20,21], AIT is contraindicated, e.g., in patients with uncontrolled and severe asthma, active systemic autoimmune disorders unresponsive to treatment, and active malignant neoplasia. Moreover, AIT should not be initiated during pregnancy, although an ongoing AIT is permissible when the patient has well tolerated it in the past. Finally, AIT is not recommended in patients with immune deficiencies, active infections and infestations, and uncontrolled diseases such as diabetes, inflammatory bowel disease, gastric ulcer, etc. It is important to note that there is no consensus on defying absolute and relative contraindications among different national and international societies.

According to an EAACI position paper on clinical contraindications to allergen immunotherapy [67], a general lack of data, especially on specific sub-classes of patients, subsists, and it might partially steer some of the aforementioned recommendations. Four representative examples will be concisely discussed; a higher risk exists in prescribing AIT to patients with uncontrolled asthma [68], but there is an essential statistical, methodological, and clinical heterogeneity in studies of AIT in asthma, and patients with severe asthma are generally excluded from clinical trials [67]. No controlled studies on the risks associated with AIT in allergic patients with underlying autoimmune diseases can be found in the literature; therefore, concern about the worsening of the underlying disease is largely hypothetical and the contraindications are suggested due to a lack of data. The increased risk of a disease exacerbation due to AIT in patients with tumors is also theoretical [69] and this contraindication has been established for the safety. Finally, according to a recent review [70], the continuation of AIT during pregnancy appears safe and the few data available suggest that the initiation of AIT during pregnancy might also be safe. Previous considerations and examples are needed to understand the urgency of real-life data in broadening the pool of possible patients eligible for therapy.

As summarized in the EAACI Guidelines on AIT in house dust mite-driven allergic asthma [21], there is an important gap in evidence for a wide group of patients who are generally excluded from RTCs for various reasons. These patients include people with multiple comorbidities, people who smoke, people with obesity, the elderly, people with suboptimal lung function, etc., who represent more plausible real-life patients than the ones selected in RTCs. The same document indicates real-life studies as suitable plans to address these open questions.

Moreover, the recent COVID-19 pandemic has put the security of AIT to the test in an unprecedented and non-replicable context: data collected from an EAACI survey made available in July 2020 on practical aspects in AIT routine and specific tolerability under the COVID-19 showed no concerns regarding a reduced tolerability under these real-life circumstances [71,72].

Due to the short observational period characteristic of RCT, there is also a need for more data about the long-term efficacy of AIT after cessation; long-lasting real-life studies and registries might fill this gap.

Furthermore, we believe that collecting homogeneous data on both elderly and children in the same pool might not only provide a novel insight into the peculiarity of the two types of patients but might allow an easier comparison between the two, easily enlightening differences and similarities. Considering the high plasticity of the immune system in children, with a possible enhanced tolerance time-window during childhood, RIAIT will allow to follow AIT-treated children over time until adulthood and identify the benefit of an early disease-modifying intervention [73].

Adherence to AIT needs to be addressed as well; there is a great variance between studies in the adherence rates, with results usually poor and largely improvable [11,74,75,76]. The range of reported adherence varied from 18% to over 90%, higher in clinical studies than in real-life surveys. In this scenario, a registry might provide novel data on adherence (by means of indirect assessment such as the following: percentages of dropouts, number of pack prescriptions compared to the prescribed dosage, poor adherence reported by the patient himself/herself, etc.) and possible relationships with characteristics, e.g., specific products and adverse effects, as well as insightful information on the reasons for the treatment interruption that might serve as a basis to develop potential ways to improve compliance.

Another hot topic is the need to find reliable and consistent criteria for AIT cessation, either after the occurrence of an adverse reaction or because the therapy is not providing the anticipated outcome. For instance, according to the EAACI Guidelines in house dust mite-driven allergic asthma [21], it should be evaluated after one year of AIT. However, currently, there is no consensus on efficacy criteria specific to allergic asthma. There is no evidence to allow any recommendations, e.g., on a shift to another product, neither about the route of administration, protocol of desensitization, nor company specific preparations. Moreover, many authors reported the absence in the literature of biomarkers that sufficiently predict a response to AIT that can be used to decide on the initiation or cessation of AIT [77]. In a recent study, Caruso et al. [78] showed that clinical biomarkers were confirmed to be helpful in monitoring AIT efficacy. As for laboratory biomarkers, a BAT (basophil activation test) showed a reduction trend, particularly for D2C1 (HDM allergen), suggesting that this is a useful parameter in monitoring patients. Finally, the need for better-designed, harmonized, and validated clinical outcomes to be used in studies has been inquired.

The RIAIT registry might help partially fill these critical gaps; the consistent study and re-evaluation of a large and heterogeneous group of patients would possibly provide new knowledge on specific characteristics related to differences in response to the treatment, as well as enlighten the most reliable clinical, instrumental, and laboratory biomarkers and suggest which clinical outcome is better among the currently used, if not suggesting new and unsuspected ones.

Another topic of increasing relevance is the potential combination of biologics with specific immunotherapy, such as omalizumab. Various trials have been conducted in recent years exploring omalizumab’s effect on the tolerability of specific immunotherapy and possible positive interactions between the two treatments [79] (recently reviewed by Dantzer et al. [80]), and the following overall positive results have been observed: AIT-biological combination treatment seems to realize a complex multitargeted treatment strategy allowing for consistently improving disease control [81]. Although GINA recommendations [22] suggest AIT for specific patients and a large part of the document is focused on biologic management in severe asthma, the combination of the two might be mentioned. As the use of biological treatment becomes more common and the number of molecules approved for severe asthma constantly increases, it will soon be clinically essential to have sufficient data to associate the effect of the two strategies. A recent study has been published on the efficacy of the combination of AIT as an add-on dupilumab in asthma with rhinitis [82], further highlighting the urgent need for data.

Continuing in the footsteps of the general review of updated evidence on the efficacy of AIT, especially in asthma [22,45], questions on the sustainability and general cost-effectiveness of this therapy have recently arisen, despite the fact that past pharmacoeconomic studies and cost-utility analyses have generally reported that AIT is economically advantageous [20]. The already mentioned review of Dhami S. et al. [45] took into consideration three studies as the best conducted on this topic and all three demonstrated an economic advantage in treating allergic asthmatic patients with AIT [83,84,85]; nevertheless, even these three studies were not without flaws and yet again there is a lack of robust long-term information in different social-economic settings. The discussion on the economic sustainability of AIT and its cost-effectiveness ratio is still open [86].

Finally, there is a problem with standardizing allergen extracts and formulations. For the common allergens, many companies provide characterized, standardized, and stable preparation for AIT. However, there is also an urgent need for a thorough harmonization of international standards in AIT [82] due to the highly heterogeneous regulation across different countries [87]. Indeed, in non-standardized preparations, the exact amount of the allergen extract might vary between batches [88,89]. In fact, some products might benefit from appropriate studies [90]. RIAIT, confronting different available products and registering their outcomes on heterogeneous patients, might provide clues on the quality of some of these formulations and help measure their impact on the patient’s disease.

Certain limitations of RIAIT should be acknowledged. Firstly, as already summarized, the quality of a registry depends on the quality of the data it contains (e.g., what is measured, in whom, how, and the extent of missing data). Secondly, due to their design, registry data might possess intrinsic lower internal validity than data collected in RCTs, limiting the extent to which they can support causal relationships.

## 4. Conclusions

Data on the long-term efficacy and safety of AIT in allergic respiratory diseases cannot be obtained using RCTs alone, and real-world studies designed as registries might serve as a complementary source of data to better define the efficacy and effectiveness of this therapy in a real-life scenario.

RIAIT presents an excellent opportunity to improve the quality of care for AIT patients with allergic respiratory diseases in Italy. A national multicentric prospective national registry will provide a large amount of data to better understand the clinical effectiveness, efficacy, and safety of real-world use of these expensive treatments.

The importance of the registry will be supported by the fact that it will be possible to reconsider contraindications critically, collect data on adult and pediatric patients, including the ones with multiple comorbidities usually neglected by RCTs, assess adherence, facilitate interoperability, data sharing, cross-comparison, and standardization among centers, and evaluate data on clinical short-term and long-term efficacy and safety, disease-modifying effects, criteria for AIT cessation, more reliable biomarker and rational clinical outcomes, and potentially clinical remission.

## Figures and Tables

**Figure 1 jpm-14-00854-f001:**
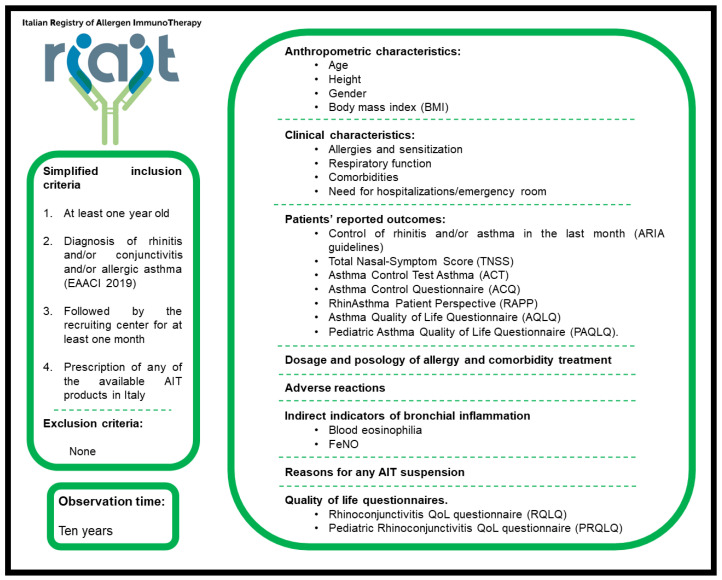
Inclusion criteria, observation time, and data examples of data collected.

## Data Availability

Not applicable.

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
