# Peer review of "RIAIT (Italian Registry of Allergen Immunotherapy): Protocol for a New Tool in a New Vision of Disease-Modifying Therapy for Allergists"

_jpm, 2024, doi:10.3390/jpm14080854_

Round 1

Reviewer 1 Report

Comments and Suggestions for Authors

This is a very good effort for a 10-year-registry with hopefully a high impact result facilitating other ongoing researches on immunotherapy and potential biomarkers. It is a very extensive data registry. Suggest data collection to include (1) allergy test such as skin prick test besides serum specific IgE test; (2) the side effects of immunotherapy. 

Author Response

We thank the Reviewer for his/her appreciation of our manuscript.

Here you can find our point-by-point response to your comments:

REVIEWER'S COMMENT:  Suggest data collection to include (1) allergy test such as skin prick test besides serum specific IgE test;

AUTHORS' RESPONSE: Thanks for your valuable comment. In the revised version of the manuscript we clarified that "The confirmation of allergenic sensitization must be documented by skin prick tests and/or serum specific IgE" (see "2.2.1. Inclusion and exclusion criteria") and that collected data about allergen sensitizations will be "assessed by means of skin prick tests and/or serum specific IgE" (see "2.2.2. Data Collection")

REVIEWER'S COMMENT: Suggest data collection to include (2) the side effects of immunotherapy

AUTHORS' RESPONSE:  Actually, for brevity this aspect was omitted in the description of the protocol, but we agree with the Reviewer that it is a very important piece of data which is also collected in the eCRF. In the revised version of the manuscript we specified that we will collect data on adverse events (see "2.2.2. Data Collection", last bullet point)

Reviewer 2 Report

Comments and Suggestions for Authors

A real-life data base national registry for AIT for allergic diseases is the need for the hour

A standardized data collection and a follow-up for 10 years is a major highlight

In the protocol

There is a mention of FENO but not spirometry.

For asthma severity, ACT can be added

Where possible, the centers can further phenotype asthma and AR 

Adherence is mentioned, but how is it measured? Details on adherence to AIT can be added

Statistical analysis uses basic tools. Please see if advanced modeling can be used to identify responders and non-responders (GEE, GLM)

would a central committee monitor the quality of data collection and give feedback to centers not meeting data quality criteria?

Comments on the Quality of English Language

Minor grammatical and spelling errors

Author Response

We thank the Reviewer for his/her appreciation of our manuscript.

Here you can find our point-by-point response to your comments:

REVIEWER'S COMMENT: There is a mention of FENO but not spirometry.

AUTHORS' RESPONSE: Actually, for brevity this aspect was omitted in the description of the protocol, but we agree with the Reviewer that it is a very important piece of data which is also collected in the eCRF. In the revised version of the manuscript we specified that we will also collect data relating to lung function (see "2.2.2. Data collection", second-last bullet point)

REVIEWER'S COMMENT: For asthma severity, ACT can be added

AUTHORS' RESPONSE: As reported in the Patients’ reported outcomes (PROs) bullet point of "2.2.2. Data collection", ACT will be collected

REVIEWER'S COMMENT: Where possible, the centers can further phenotype asthma and AR 

AUTHOR'S RESPONSE: We thank the Reviewer for this very valuable comment. We agree with him/her that, if centers will have the opportunity to do it, further phenotypization for both asthma and allergic rhinitis will be very informative. That's the why in the eCRF we included also the collection of data about biomarkers. We will consider, out of the eCRF, also additional data that will come from centers available to further phenotyping patients (see the revised version of the manuscript, "2.2.2 Data collection", last paragraph).

REVIEWER'S COMMENT: Adherence is mentioned, but how is it measured? Details on adherence to AIT can be added

AUTHOR'S RESPONSE: We really thank the Reviewer for this important comment. Unfortunately, no standardized and validated methods are available to assess AIT adherence; however, we believe that by means of a registry (such as RIAIT) adherence could be indirectly assessed (i.e.: evaluating percentages of dropouts, number of pack prescriptions compared to the prescribed dosage, poor adherence reported by the patient himself/herself...). We added these details in the revised Discussion ("3.1. Addressing open questions and unmet needs", 7th paragraph).

REVIEWER'S COMMENT: Statistical analysis uses basic tools. Please see if advanced modeling can be used to identify responders and non-responders (GEE, GLM)

AUTHORS' RESPONSE: Thanks for this suggestion that we will take in consideration when performing statistical analysis on the collected data. We updated the statistical analysis paragraph accordingly (see "2.2.5. Duration of the study and statistics")

REVIEWER'S COMMENT: would a central committee monitor the quality of data collection and give feedback to centers not meeting data quality criteria?

AUTHORS' RESPONSE: Yes, as stated in "2.2.5. Duration of the study and statistics", 5th paragraph: "A medical data review will also be carried out once a year. These analyses allow for the identification of any corrective interventions aimed at improving the quantity and quality of the information collected among those provided in the CRF; these suggestions will be communicated to the manager of the individual center."

Reviewer 3 Report

Comments and Suggestions for Authors

General comment

This article describes a protocol for a prospective multicenter registry for adults and children starting AIT. This mass of information might gain sufficient statistical power to address important clinical and research questions, and a national registry will minimize the variability of data collected by standardizing variables across geographical regions, thus enhancing our understanding of the allergic population by examining the response to AIT based on characteristics like phenotypes, biomarkers, concomitant treatments, and socio-economic status. RIAIT (Italian Registry of Allergy Immunotherapy) is a multicenter observational study that aims to collect the most extensive data available on Italian patients using AIT for respiratory allergies (rhinoconjunctivitis and/or asthma) in a real-life context, through a network of allergy reference centers with specific interest and expertise in the management of allergic respiratory patients. I think, RIAIT presents an excellent opportunity to improve the quality of care for AIT patients with allergic respiratory diseases in Italy.

Specific points

1- The authors should not use abbreviations at the beginning of sentences and lines (e.g.)

- RIAIT is promoted by the Italian Society of Allergy, Asthma and Clinical Immunology (SIAAIC)[26] and the Italian Society of Pediatric Allergy and Clinical Immunology (SIAIP)[27].

Author Response

We thank the Reviewer for his/her appreciation of our manuscript.

Here you can find our point-by-point response to your comments:

REVIEWER' COMMENT: The authors should not use abbreviations at the beginning of sentences and lines 

AUTHORS' RESPONSE: In the revised version of the manuscript we corrected accordingly.